# Application of Novel Pharmacists’ Risk in Pharmacotherapy (PHARIPH) Scale for Identification of Factors Affecting the Safety of Hospital Pharmacotherapy—An Observational Pilot Study

**DOI:** 10.3390/ijerph19031337

**Published:** 2022-01-25

**Authors:** Olga Fedorowicz, Łukasz Rypicz, Anna Wiela-Hojeńska, Ewa Jaźwińska-Tarnawska, Izabela Witczak

**Affiliations:** 1Department of Clinical Pharmacology, Wroclaw Medical University, 50-367 Wroclaw, Poland; anna.wiela-hojenska@umw.edu.pl (A.W.-H.); tje1@wp.pl (E.J.-T.); 2Clinical Pharmacy Service, University Teaching Hospital, 50-556 Wroclaw, Poland; 3Department of Population Health, Wroclaw Medical University, 50-367 Wroclaw, Poland; lukasz.rypicz@umw.edu.pl (Ł.R.); izabela.witczak@umw.edu.pl (I.W.)

**Keywords:** pharmacist, risk factors, safe pharmacotherapy

## Abstract

Background: The widespread occurrence of medication errors (MEs) has become a global problem because it poses a serious threat to the health and lives of patients, can prevent the achievement of treatment goals, undermines patient trust in the health care system, and increases treatment costs. The purpose of this study was to develop an appropriate tool to identify key risk factors that hospital pharmacists believe threaten pharmacotherapy safety in the hospital. Methods: A diagnostic survey method using the authors’ PHARIPH (Pharmacists’ Risk in Pharmacotherapy) scale and authorial questions was used to identify risks that may result in patient pharmacotherapy errors at the hospital pharmacist level. A total of 125 Polish hospital pharmacists participated in the study. Results: The original authors’ created PHARIPH scale was characterized by a Cronbach’s alpha coefficient of 0.958. According to the surveyed pharmacists, the greatest threat to pharmacotherapy safety was misreading of a doctor’s order (similar drug nomenclature) and preparing a wrong drug (similar drug packaging, similar drug nomenclature). Female pharmacists compared to male pharmacists attributed significantly higher importance to such risk factors such as pharmacist’s ignorance of a list of drug substitutes (*p* = 0.047, risk 8), preparation from an expired/withdrawn drug (*p* = 0.002, risk 14), preparation from a drug stored in inappropriate conditions (*p* = 0.05, risk 15), preparation of drugs ordered in hospital and PODs (patients’ own drugs) without checking for possible drug duplication (*p* = 0.011, risk 17) and their potential effect on patient safety. Conclusions: The PHARIPH scale could be applied as a novel tool for identification of pharmacotherapy risks.

## 1. Introduction

Pharmacotherapy is a process that comprises several steps, including diagnosis, prescribing, properly preparing and dispensing the right drug, administering it to the patient, and monitoring its effects of action [1,2]. In modern medicine, pharmacotherapy is considered the most common method of treatment and, when used properly, it contributes to significant health improvement in patients. The use of medication, however, may also be a source of errors that occur more frequently in some healthcare areas than in others [2]. The common occurrence of errors in the pharmacotherapy process has become a global issue, as it constitutes a serious threat to achieving therapeutic purposes, undermines patient confidence in the healthcare system, and increases treatment costs. Treatment errors are among the major factors that contribute to patient morbidity and mortality [3].

The errors that occur in the pharmacotherapy process and cause (or have the potential to cause) harm to the patient are called medication errors (ME) [4,5]. To prevent such incidents from occurring in the future, it is necessary to analyze them to discover the causes and determine the proper procedure. The available strategies should make it possible to prevent the recurrence of errors both at the level of the organization, namely the hospital, and at the level of the professional activity of individual health professionals [6]. Their preparation is extremely important because many medical professionals, including doctors, nurses, and pharmacists, are involved in the implementation of individual stages of pharmacotherapy. The proper communication among these groups may considerably reduce medical errors [6,7]. An analysis conducted in the United Kingdom (2019) showed that the annual number of ME amounted to approximately 237 million, and they occurred at all stages of the process of medication use: prescribing (21.3%), transition (1.4%), dispensing (15.9%), administering (54.4%), and monitoring (7.0%) [8]. A Canadian study based on a review of the 2013–2017 literature results found that a quarter of approximately 70,000 adverse events reported annually were related to treatment errors that caused almost 700 deaths per year. Each preventable drug-related adverse event in a hospitalized patient (in-patient) costs approximately 6750 Canadian dollars (inflation adjustment to 2017) and prolongs the patient’s hospitalization by 4.6 days [9].

Pharmacists form a professional group that can play a key role in the prevention and detection of medication errors [10,11,12]. In the Polish healthcare system, in contrast to many others in Europe and worldwide, the role of a pharmacist (especially in hospitals) is very limited. The tasks of a Polish pharmacist mainly include the routine provision of information to the medical personnel about properties of a drug, its adverse effects, potential drug interactions, or availability of medications in the hospital [13]. Therefore, hospital pharmacists should adopt a new approach to the promotion of safe medication use and implement practices based on risk-management principles [14].

This article aims to specify pharmacotherapy safety (pharmacovigilance) at the hospital pharmacist level by identifying key risk factors that may affect safe administration of drugs to in-patients. The purpose of this study was to develop an appropriate tool to identify key risk factors that hospital pharmacists believe threaten pharmacotherapy safety in the hospital.

Designing an original research tool to assess pharmacotherapy risk factors, PHARIPH (Pharmacists’ Risk in Pharmacotherapy) scale, is a novel approach to the issue of treatment safety. The topic is very timely and fits into the patient safety policy announced by the WHO. In the available literature there is no information about a similar tool.

## 2. Materials and Methods

This study is a part of a larger project that assessed the relevance of individual drug-related risk factors that trigger pharmacotherapy errors (hereafter referred to as risk), which may affect safe administration of medications to inpatients. The study survey questionnaire was available on the website www.webankieta.pl (between 1 May 2019 to 31 August 2019). A link to the tool was sent to the email addresses of hospital pharmacists. The inclusion criterion was a master’s degree in pharmacy and employment in healthcare facilities providing 24 h services, i.e., hospital pharmacies or hospital pharmacy departments throughout Poland. Furthermore, the study was promoted at conferences dedicated to hospital pharmacists. Each interested person anonymously and electronically completed the questionnaire. After the questionnaire had been completed, the respondent validated and uploaded it to the online platform, where the data were collected. A total of 125 Polish hospital pharmacists participated in the study (in Poland there are 949 hospitals and 1700 hospital pharmacists).

### 2.1. Pharmacists’ Risk in Pharmacotherapy (PHARIPH) Scale

The authorial PHARIPH tool, comprising the socio-demographic particulars of the respondents and a three-part risk scale to assess selected risk triggers in the pharmacotherapy process, was applied in the study. This article shows one part of the matrix, dedicated to hospital pharmacists, with seventeen risk factors (Table 1).

The reliability of the scale used for assessing the responses obtained was tested by calculating the Cronbach’s alpha coefficient and discriminatory power for individual risks and the Cronbach’s alpha coefficient for the whole scale. The coefficient adopts values from 0 to 1. The larger the value, the greater the scale reliability. Values above 0.7 are deemed to indicate standard scale reliability [15].

The Cronbach’s alpha for the authors’ PHARIPH scale was 0.958. All risks included in the scale had positive discriminatory power. This means that they positively correlated with other risks included in the scale, which was a very desirable effect. The results also indicated that exclusion of any of the risks did not increase the Cronbach’s alpha coefficient. Based on the above-mentioned parameters, it can be concluded that the scale applied in the study was reliable and the results of its use were reproducible.

The materiality of the occurrence of individual risks that affect the patient’s pharmacotherapy process was determined using a five-point scale, where 1 means negligible risk, and 5 a very significant risk. The measure of the materiality of risk was the mean of scores (M) obtained in the questionnaire for each risk, ranging from 1 to 5. The higher the mean, the greater the importance of a given risk to the pharmacotherapy process. The authors assumed that a mean ranging from 1.00 to 1.79 for an individual risk indicated negligible materiality, from 1.80 to 2.59 insignificant, from 2.60 to 3.39 significant, and from 3.40 to 4.19 quite significant, and the mean above 4.20 showed very significant materiality for the pharmacotherapy process.

### 2.2. Statistical Methods

The database and preliminary analysis of the results were performed using Microsoft Excel. An in-depth statistical analysis was conducted using Statistica 13.3 software. Nonparametric tests, including the Mann–Whitney U test and Kruskal–Wallis test with a possible post hoc analysis (Dunn’s test), were used for statistical calculations. Test values meeting the *p* < 0.05 condition were deemed statistically significant. The analysis was conducted using R-3.6.1 software (R Core Team, 2019).

### 2.3. Ethical Considerations

The study was fully anonymous and voluntary. The research project was approved by the independent Bioethics Committee at the Wroclaw Medical University (No KB–610/2017). All participants granted their written informed consent after a thorough explanation of the procedures involved. The study was carried out in accordance with the tenets of the Declaration of Helsinki and guidelines of Good Clinical Practices (World Medical Association, Ferney-Voltaire, France, 2013).

## 3. Results

### 3.1. Study Group

A total of 125 pharmacists, who completed the questionnaire and uploaded the survey results to the online platform, participated in the study. The characteristics of the study group are detailed in Table 2. In Poland, according to the current legislation, the manager of a hospital pharmacy can be a pharmacist who meets the following conditions: has at least two years of full-time experience in the practice of the profession in a hospital pharmacy and holds a specialist title in hospital pharmacy, clinical pharmacy, or retail pharmacy.

### 3.2. PHARIPH Scale Analysis

The mean calculated from the scores obtained from the survey according to the 5-point scale for individual risks showed that the surveyed pharmacists attributed the highest materiality to the risk of misreading of a doctor’s order (similar drug nomenclature) and preparation of a wrong drug (similar drug packaging, similar drug nomenclature) (risks 1 and 2). The mean (M) of the scores obtained from the responses (4.22) indicates that the materiality of those factors was assessed as “very significant”. Risks such as preparation of medication in the wrong dose (M = 4.11), preparation of both medications ordered in hospital and patient’s own drugs (M = 4.11), and work under time pressure during drug preparation (M = 4.02) were assessed as “quite significant” for the pharmacotherapy process (risks 3, 17, and 6, respectively).

The remaining factors were rated at slightly below 4 points, which means that they were also “quite significant.” The responses to questions 4 and 5, for which the risk was rated around 3.35 points (“significant”) were the exceptions (Table 3).

The study revealed statistically significant relationships between sex, age, and job seniority of the surveyed pharmacists (Table 4).

Women rated the materiality of the following risk factors significantly higher than men:Pharmacist’s ignorance of a list of drug substitutes (risk 8);Preparation of a pharmaceutical formulation from an expired/withdrawn drug (risk 14);Preparation of a pharmaceutical formulation from a drug stored in abnormal conditions (risk 15);Preparation of medications ordered in hospital and patient’s own drugs without verification of possible drug duplication (risk 17) and their potential effect on patient safety.

The pharmacy personnel aged 20–29 significantly higher rated the importance of not having an online ordering system than personnel aged 30–59 (risk 12; Table 5).

The pharmacists with seniority of up to 5 or more than 30 years rated significantly higher the risk of replacing existing medications with others due to market shortages (risk 5) and preparation of a pharmaceutical formulation under inappropriate conditions (such as failure to maintain aseptic conditions) (risk 15 and 16) than the pharmacists with 11–19 years of work experience (Table 6).

The analysis did not show statistically significant relationships among questions from PHARIPH dedicated to place of residence, place of work, or additional employment of the respondents.

## 4. Discussion

Considering the patient safety as well as the safety of medical personnel (professional liability), the authors of the study attempted to define the priority risks that may affect pharmacovigilance, the source of which may be the pharmacy personnel working in the hospital.

The errors related to the pharmacotherapy process are frequently multifactorial in nature. In exceptional situations, a combination of errors might occur. Such situations can be life-threatening and health-threatening for the patient [16]. Therefore, it is of great importance to identify errors and understand their etiology to prevent them in the future.

Our own research, based on analysis of the authors’ PHARIPH scale, shows that the greatest threat to pharmacovigilance is misreading of a doctor’s order by a pharmacist due to similar drug nomenclature, both look-alike and sound-alike (LASA), and preparation of the wrong drug, which results from the similarity of drug nomenclature and drug packaging. The surveyed pharmacists assessed that those risk factors were among the most significant ones in their work (M = 4.22). The problem of LASA drugs has been known for decades. Teplitsky described that issue back in the 1970s [17]. He pointed out that similar drug nomenclature could be a particularly serious threat to pharmacovigilance in the case of verbal telephone orders, but also those that were handwritten by doctors. An illegible doctor’s order, which has not been timely disputed and returned for correction, can carry risks to the patient, sometimes fatal [18,19,20]. According to Trbovich and Hyland, with more than 20,000 drug trade names in the Canadian market, the problem of LASA drugs poses a serious challenge to healthcare professionals. They also stressed the great financial risk resulting from LASA drugs, as the costs of nullifying such a risk affecting patient safety increase exponentially when the risk is identified after the drug is on the market [21].

According to our study, when preparing a dose of a drug at the pharmacy unit, there is a high risk (M = 4.11) of preparing a wrong dose, which is the effect of LASA drugs. The pharmacy personnel indicated that identical graphic design of packaging for different doses and poor labelling of the drug dose on the packaging were significant factors that could cause such an error. Usually, drug concentration (expressed, for example, as the amount of active substance in 1 mL), is highlighted on packaging by the use of capital letters, while drug capacity (expressed in mL) is barely visible (lower case).

The factors assessed as quite significant by the surveyed pharmacy personnel (PHARIPH scale), predisposing to drug dispensing errors or errors related to the preparation of drugs, also included performance of work under time pressure during the preparation of drugs. That results, among other issues, from too late doctor’s orders, staffing shortages in pharmacies, waiting for a drug from the wholesaler or for an administrative decision concerning “expensive drugs” dispensed with special permission of the Medical Director (M = 4.02).

Similar problems associated with the duties of pharmacists were highlighted by *The Royal Pharmaceutical Society of Great Britain* (RPSGB) in a report of a symposium held in 2009. The report identified the following causes of errors: lack of rest breaks at work, staffing shortages, and excessive workload [22]. Similarly, Tsao et al. in their study involving pharmacists in a Canadian province identified insufficient rest breaks at work and limited time to perform specific activities as possible reasons for committing an error [23]. In contrast, the results of a study conducted in China prove that the stress accompanying the professional tasks of a hospital pharmacist results in insomnia, the desire to reduce working hours, to resign from the job, or to change the tasks performed as part of professional duties [24].

The presented studies and our own results show that excessive workload and pharmacy personnel shortage in hospitals adversely affect pharmacovigilance. The number of hospital pharmacists in individual healthcare systems worldwide varies widely. The Netherlands is an example of the shortage of pharmacy personnel. As few as 0.75 hospital pharmacists on average are available per 100 hospital beds compared to 1.42 in the UK and 14.1 in the US [25]. In Poland, on average, there is usually one hospital pharmacist for an entire hospital. Only larger institutions (provincial and university hospitals) employ several or more pharmacists [26].

The pharmacists surveyed in our study also identified the problem associated with the supervision of *patients’ own drugs* (PODs) as quite significant. On the one hand, medications brought by the patient make it possible to recreate the list of medications taken by the patient, which improves safety. However, on the other hand, they can cause errors if the patient takes them on their own without the knowledge of medical personnel. It may also happen that the medical personnel administer PODs to the patient and simultaneously, by mistake, also the same drug ordered by a doctor to the hospital pharmacy. The Clinical Pharmacy Standards in the United Kingdom, where a clinical pharmacist verifies drugs and dietary supplements taken by the patient within 24 h of the patient’s admission to hospital (medication conciliation process), can be an example of good practice for the supervision of PODs in hospitals [27]. In Poland, in hospitals undergoing accreditation processes, this problem should be solved by establishing and implementing a procedure to be applied to PODs. PODs should be registered and secured by medical personnel. Furthermore, a patient should sign a declaration confirming that they will not self-administer any medications, over-the-counter (OTC) drugs, or dietary supplements during their hospitalization. Unfortunately, practice shows that patients do not always follow these rules. This may result in ineffective therapy and many possible side effects, including those classified as severe. Grissinger analyzed reports concerning medication errors involving patients taking their own drugs in hospital, which were reported to the *Pennsylvania Patient Safety Authority* (USA) between 1 July 2004 and 31 January 2011. Out of 879 errors reported, most were situations in which in-patients self-administered medications without informing facility staff [28]. There is little in the literature on the frequency with which patients bring their own medications to the hospital. In the study by Nielsen et al., which involved 529 patients, it was observed that more than half of the patients (59%) brought their own drugs to hospital; however, only 7.5% of those patients self-administered them. This means that in most cases the replacement of PODs with drugs from the Hospital Formulary was preferred [29]. To ensure patient safety in this regard, information concerning institution policy regarding the use of PODs during hospitalization should be provided to both the patient and their family prior to admitting the patient to the hospital.

In our study (based on application of the PHARIPH scale) revealed statistically significant relationships between sex, age, and job seniority of the surveyed pharmacists and the importance of specific risk for occurrence of medical errors. The women significantly higher than man rated the materiality of the risk factor such as preparation of a pharmaceutical formulation from an expired/withdrawn drug. An important factor in determining whether a medication is safe to use and will work as intended is the expiration date of the medication. As defined by the World Health Organization (WHO), expiration date means: “the date indicated on the individual packaging of a pharmaceutical product, which includes the date on which the product is expected to remain within its specified performance when properly stored for each batch by adding the shelf life to the manufacturing date” [30]. The use of expired medicinal products is risky and potentially harmful to health, especially in the case of biological drugs, vaccines, or antibiotics.

If a drug has degraded, it might not provide the patient with the intended benefit because it has a lower strength than intended. In addition, when a drug degrades, it may yield toxic compounds that could cause consumers to experience unintended side effects. In order to prevent the use of drugs after their expiration date, hospital pharmacists create procedures for the correct rotation of drugs in the hospital pharmacy and in wards and for checking the expiration date before the drug is prepared. In addition to the safety of therapy, expired medications are also an economic and environmental problem [31].

Pharmacists have also drawn attention to generic drugs, and their unfamiliarity with them can cause errors. Hospital pharmacists tend to select the most cost-effective generic drug available on the market when bidding for drugs, which means that frequently used generics will periodically change from one brand (generic or innovator) to another. This makes their prescribing process more complex and can have a large impact on clinical outcomes. This is particularly important for drugs with a small therapeutic window, where even small differences in bioequivalence can cause the failure of the therapy or the occurrence of adverse effects [32].

### Limitations of the Study

This study was subject to the inherent limitations of survey-based methods, including the pharmacist’s bias. It was not possible to verify the truthfulness of the participants’ answers; therefore, the validity of our results may be limited. The population of participants was not a randomized selection, and the number of participants was relatively small, so results should be interpreted with caution. More research is needed to confirm such observations.

## 5. Conclusions

PHARIPH scale could be applied as a novel tool for identification of pharmacotherapy risks.

This study has shown that the significant risk factors in the pharmacist’ duties, identified by PHARIPH scale, which may affect pharmacovigilance, include illegible doctor’s orders, similar drug packaging and drug nomenclature, unclear specification on packaging in terms of the extent of the drug effect, and patients’ own drugs (PODs). All these factors may be the cause of preparing a wrong drug, a wrong dose, or drug duplication. The work under time pressure, which is caused, among other things, by doctor’s orders issued too late, staffing shortages, prolonged waiting for a drug from the wholesaler, or for an administrative decision on the possibility of using a given drug, is also a significant element resulting in the occurrence of errors. Relevance of these risks has been identified, which will enable the proper procedures to be developed to prevent serious errors in the future.

## Figures and Tables

**Table 1 ijerph-19-01337-t001:** PHARIPH, part three of the survey—risk triggers for hospital pharmacists.

Risk	Risk Factor	Cronbach’s Alpha after Risk Exclusion	Discriminatory Power
1	Misreading of a doctor’s order (similar drug nomenclature)	0.955	0.796
2	Preparation of a wrong drug (similar drug packaging, similar drug nomenclature)	0.954	0.809
3	Preparation of a medication in a wrong dose (drug concentration highlighted on packaging vs. barely visible drug capacity)	0.955	0.792
4	Frequent changes in trade drug names in a hospital, e.g., due to a new tender	0.959	0.513
5	The need to replace existing medications with new ones due to shortages in the market	0.958	0.56
6	Time pressure during drug preparation due to, among other things, late orders of a doctor, low staffing, waiting for a medication from a wholesaler or for an administrative decision	0.957	0.653
7	Improper work organization (e.g., answering phone calls, performing other tasks “in the meantime”)	0.957	0.633
8	Pharmacist’s ignorance of a list of drug substitutes	0.954	0.835
9	Errors in doctor’s orders that were unnoticed by the pharmacist before preparation of the drug	0.954	0.866
10	Psychophysiological fatigue	0.956	0.726
11	Scarce availability of training concerning drug preparation.	0.958	0.617
12	No online ordering system	0.958	0.593
13	Ignorance of drug preparation procedures	0.954	0.815
14	Preparation of a pharmaceutical formulationfrom an expired/withdrawn drug	0.954	0.847
15	Preparation of a pharmaceutical formulation stored in improper conditions	0.954	0.851
16	Preparation of a pharmaceutical formulation under inadequate conditions, such as failure to maintain aseptic conditions	0.954	0.846
17	Preparation of medications ordered in hospital and concomitant patient’s self-administration of own drugs without the knowledge of medical personnel	0.954	0.825

**Table 2 ijerph-19-01337-t002:** Sociodemographic data of the study group.

Factor	Category	n = 125	% of Total Participants
sex	female	93	74.4
	male	32	25.6
age (years)	20–29	10	8.0
	30–39	44	35.2
	40–49	31	24.8
	50–59	29	23.2
	65 and over	11	8.8
job seniority (years)	up to 5	14	11.2
	6–10	21	16.8
	11–19	40	32.0
	20–29	29	23.2
	30 and over	21	16.8
specialization	retail pharmacy	42	33.6
	clinical pharmacy	14	11.2
	hospital pharmacy	33	26.4
	Others	8	6.4
size of the town where he/she works	city up to 50 thousand inhabitants	32	25.6
	city between 50 thousand and 100 thousand inhabitants	11	8.8
	city between 100 thousand and 500 thousand inhabitants	21	16.8
	city with more than 500 thousand inhabitants	61	48.8

**Table 3 ijerph-19-01337-t003:** The importance attributed by the surveyed pharmacists to individual risks (survey questions) in pharmacotherapy.

Risk	Very Significant (5)	Quite Significant (4)	Significant (3)	Insignificant (2)	Negligible (1)	M
1	68.00%	8.00%	10.40%	4.80%	8.80%	4.22
	N = 85	N = 10	N = 13	N = 6	N = 11	
2	68.00%	9.60%	7.20%	7.20%	8.00%	4.22
	N = 85	N = 12	N = 9	N = 9	N = 10	
3	56.00%	18.40%	12.80%	6.40%	6.40%	4.11
	N = 70	N = 23	N = 16	N = 8	N = 8	
4	18.40%	25.60%	33.60%	15.20%	7.20%	3.33
	N = 23	N = 32	N = 42	N = 19	N = 9	
5	16.80%	24.00%	43.20%	10.40%	5.60%	3.36
	N = 21	N = 30	N = 54	N = 13	N = 7	
6	42.40%	23.20%	28.80%	4.80%	0.80%	4.02
	N = 53	N = 29	N = 36	N = 6	N = 1	
7	39.20%	24.80%	28.80%	5.60%	1.60%	3.94
	N = 49	N = 31	N = 36	N = 7	N = 2	
8	31.20%	24.00%	21.60%	11.20%	12.00%	3.51
	N = 39	N = 30	N = 27	N = 14	N = 15	
9	47.20%	20.00%	17.60%	9.60%	5.60%	3.94
	N = 59	N = 25	N = 22	N = 12	N = 7	
10	40.00%	22.40%	32.80%	3.20%	1.60%	3.96
	N = 50	N = 28	N = 41	N = 4	N = 2	
11	28.80%	27.20%	28.80%	10.40%	4.80%	3.65
	N = 36	N = 34	N = 36	N = 13	N = 6	
12	25.60%	28.80%	27.20%	9.60%	8.80%	3.53
	N = 32	N = 36	N = 34	N = 12	N = 11	
13	38.40%	20.00%	22.40%	9.60%	9.60%	3.68
	N = 48	N = 25	N = 28	N = 12	N = 12	
14	58.40%	12.00%	8.80%	3.20%	17.60%	3.9
	N = 73	N = 15	N = 11	N = 4	N = 22	
15	52.80%	21.60%	4.80%	4.80%	16.00%	3.9
	N = 66	N = 27	N = 6	N = 6	N = 20	
16	58.40%	16.80%	4.80%	2.40%	17.60%	3.96
	N = 73	N = 21	N = 6	N = 3	N = 22	
17	56.00%	16.80%	16.00%	4.80%	6.40%	4.11
	N = 70	N = 21	N = 20	N = 6	N = 8	

%—the percentage of responses provided. M—the score mean calculated from the points obtained in the questionnaire according to the 5-point scale for individual risks N—number of respondents’.

**Table 4 ijerph-19-01337-t004:** The effect of respondents’ sex on the assessment of risk parameters.

Risk	Sex	Very Significant (5)	Quite Significant (4)	Significant (3)	Insignificant (2)	Negligible (1)	M	*p*
8	Female	36.56%	24.73%	15.05%	12.90%	10.75%	3.63	0.047
		N = 34	N = 23	N = 14	N = 12	N = 10		
	Male	15.62%	21.88%	40.62%	6.25%	15.62%	3.16	
		N = 5	N = 7	N = 13	N = 2	N = 5		
14	Female	66.67%	9.68%	6.45%	3.23%	13.98%	4.12	0.002
		N = 62	N = 9	N = 6	N = 3	N = 13		
	Male	34.38%	18.75%	15.62%	3.12%	28.12%	3.28	
		N = 11	N = 6	N = 5	N = 1	N = 9		
15	Female	56.99%	22.58%	3.23%	4.30%	12.90%	4.06	0.05
		N = 53	N = 21	N = 3	N = 4	N = 12		
	Male	40.62%	18.75%	9.38%	6.25%	25.00%	3.44	
		N = 13	N = 6	N = 3	N = 2	N = 8		
17	Female	63.44%	13.98%	11.83%	4.30%	6.45%	4.24	0.011
		N = 59	N = 13	N = 11	N = 4	N = 6		
	Male	34.38%	25.00%	28.12%	6.25%	6.25%	3.75	
		N = 11	N = 8	N = 9	N = 2	N = 2		

*p*—in Mann–Whitney U test (*p* < 0.05: condition were deemed statistically significant); M—the score mean calculated from the points obtained in the questionnaire according to the 5-point scale for individual risks N—number of respondents’.

**Table 5 ijerph-19-01337-t005:** The effect of respondents’ age on the assessment of risk parameters.

Risk	Age (Years)	Very Significant (5)	Quite Significant (4)	Significant (3)	Insignificant (2)	Negligible (1)	M	*p*
12	20–29 (A)	70.00%	20.00%	10.00%	0.00%	0.00%	4.6	*p* = 0.034
		N = 7	N = 2	N = 1	N = 0	N = 0		
	30–39 (B)	18.18%	27.27%	38.64%	6.82%	9.09%	3.39	A > C, D, B
		N = 8	N = 12	N = 17	N = 3	N = 4		
	40–49 (C)	32.26%	25.81%	12.90%	16.13%	12.90%	3.48	
		N = 10	N = 8	N = 4	N = 5	N = 4		
	50–59 (D)	13.79%	34.48%	34.48%	13.79%	3.45%	3.41	
		N = 4	N = 10	N = 10	N = 4	N = 1		
	60 and over (E)	27.27%	36.36%	18.18%	0.00%	18.18%	3.55	
		N = 3	N = 4	N = 2	N = 0	N = 2		

*p*—in Kruskal–Wallis test + post hoc analysis (Dunn’s test); M—the score mean calculated from the points obtained in the questionnaire according to the 5-point scale for individual risks in a given age group; N—number of respondents’ A, B, C, D, E—designation of ranges for the age (years) criterion.

**Table 6 ijerph-19-01337-t006:** The effect of respondents’ job seniority on the assessment of risk parameters.

Risk	Job Seniority (Years)	Very Significant (5)	Quite Significant (4)	Significant (3)	Insignificant (2)	Negligible (1)	M	*p*
5	up to 5 (A)	28.57%	42.86%	21.43%	7.14%	0.00%	3.93	*p* = 0.037 A, E > C, D
		N = 4	N = 6	N = 3	N = 1	N = 0		
	6–10 (B)	19.05%	14.29%	52.38%	4.76%	9.52%	3.29	
		N = 4	N = 3	N = 11	N = 1	N = 2		
	11–19 (C)	10.00%	20.00%	50.00%	17.50%	2.50%	3.17	
		N = 4	N = 8	N = 20	N = 7	N = 1		
	20–29 (D)	10.34%	20.69%	51.72%	6.90%	10.34%	3.14	
		N = 3	N = 6	N = 15	N = 2	N = 3		
	30 and over (E)	28.57%	33.33%	23.81%	9.52%	4.76%	3.71	
		N = 6	N = 7	N = 5	N = 2	N = 1		
15	up to 5 (A)	64.29%	35.71%	0.00%	0.00%	0.00%	4.64	*p* = 0.013 A, E > C
		N = 9	N = 5	N = 0	N = 0	N = 0		
	6–10 (B)	57.14%	19.05%	9.52%	4.76%	9.52%	4.1	
		N = 12	N = 4	N = 2	N = 1	N = 2		
	11–19 (C)	37.50%	22.50%	7.50%	7.50%	25.00%	3.4	
		N = 15	N = 9	N = 3	N = 3	N = 10		
	20–29 (D)	44.83%	27.59%	3.45%	3.45%	20.69%	3.72	
		N = 13	N = 8	N = 1	N = 1	N = 6		
	30 and over (E)	80.95%	4.76%	0.00%	4.76%	9.52%	4.43	
		N = 17	N = 1	N = 0	N = 1	N = 2		
16	up to 5 (A)	78.57%	21.43%	0.00%	0.00%	0.00%	4.79	*p* = 0.03 A, E > C
		N = 11	N = 3	N = 0	N = 0	N = 0		
	6–10 (B)	57.14%	19.05%	9.52%	0.00%	14.29%	4.05	
		N = 12	N = 4	N = 2	N = 0	N = 3		
	11–19 (C)	45.00%	17.50%	5.00%	5.00%	27.50%	3.48	
		N = 18	N = 7	N = 2	N = 2	N = 11		
	20–29 (D)	51.72%	20.69%	6.90%	3.45%	17.24%	3.86	
		N = 15	N = 6	N = 2	N = 1	N = 5		
	30 and over (E)	80.95%	4.76%	0.00%	0.00%	14.29%	4.38	
		N = 17	N = 1	N = 0	N = 0	N = 3		

*p*—Kruskal–Wallis test + post hoc analysis (Dunn’s test); M—the score mean calculated from the points obtained in the questionnaire according to the 5-point scale for individual risks in the group of pharmacists with specified job seniority N—number of respondents’ A, B, C, D, E—designation of ranges for the job seniority criterion.

## Data Availability

Not applicable.

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
