# Peer review of "Application of Novel Pharmacists’ Risk in Pharmacotherapy (PHARIPH) Scale for Identification of Factors Affecting the Safety of Hospital Pharmacotherapy—An Observational Pilot Study"

_ijerph, 2022, doi:10.3390/ijerph19031337_

Round 1
Reviewer 1 Report
General comments
Medication errors associated with the pharmacotherapy process are a serious problem and challenge from a public health perspective. Therefore, this article makes an important contribution to the knowledge on the subject.
However, I have some comments on the article under review.
Material and method section
- There is no description of the studied population. Please, provide at least the number of hospitals in Poland and the number of pharmacists working in these hospitals.
- How many study invitations have been sent/distributed?
- In the questionnaire, did the authors allow the possibility of no answer or to indicate that the respondent did not have an opinion?
- Sample design/sampling - does it ensure the representativeness of the study?
- The authors provide a breakdown of individual risk (line 113-116). What is the assumption of the following breakdown? Does this result from the distribution of the variable?
Results
- The authors use the mean by analysing individual items (questions) of the scale. However, they are measured on a 1-5 scale which is an categorical scale. The use of the mean for a categorical variable is unjustified.
- Have the authors analysed whether the distribution of their scales is consistent with the normal distribution? If not, the use of the mean to describe the results is unjustified.
- The authors did not use the potential of the scale they developed. They mainly analysed the individual items (questions) of the scale. Scale results are more reliable than individual items.
Conclusions
- The conclusions are far reaching. The scale developed by the authors analyses the opinions (concerns) of pharmacists, and not the causes of real mistakes.
Minor comments
- The evaluation of the reliability of the scale (line 107-108) should rather be under discussion
- When describing the significance of the results, if they are statistically significant, we use the p <0.05 notation; p <0.01; p <0.001.
- In Table 2, I suggest entering values with an accuracy of 1 decimal place in the 4th column.
- Table 3. I propose to convert to 100% cumulative bar chart.
Author Response
Dear Reviewer,
I am enclosing my responses to the comments.
Yours sincerely
Lukasz Rypicz

Reviewer 2 Report
Materials and Methods – More information on how respondents were sampled is needed.
Line 82 – What website is this referring to?
Line 83 – How were these hospital pharmacists sampled? Was the survey sent to all hospital pharmacists in Poland or just certain regions/hospitals? Was there representation from metropolitan and regional/rural areas, for example?
Line 87 – What was the response rate for the survey? Was a sample size calculation performed? What was the ideal sample size needed and was this reached?
Line 90 – How were the items for the PHARIPH tool initially selected? Was this based on previous research, literature etc? If so, this needs to be cited. Was there any piloting of the tool done? Was face and content validity assessed?
Line 113 – the authors made certain assumptions regarding the cut-off points for different levels of risk. How were these cut-offs determined? Was it done statistically or based on previous research?
Line 120 - More detail on exactly what analyses were conducted is needed. i.e. what variables were included, what associations/differences explored etc needs to be stated a priori.
Line 130 – the inclusion criteria should be described in the Methods section
Table 2 – I was confused as to why hospital pharmacists would predominantly have retail pharmacy specialisation? Perhaps this is to do with the Polish health system - more detail needed to clarify for an international audience.
Line 234 – How are these reasons known? Did participants state this or is this from previous literature? If so, cite.
Line 286 – More discussion on reasons for these differences and what this means is needed. Currently, it is just repeating the results.
The Discussion section currently lacks some important detail. Limitations of the study need to be discussed, especially the generalisability of findings given the small sample.
Practice implications and future research directions also need to be discussed i.e. how would this tool actually be used in clinical practice? At the moment, it is a little unclear to me. Given the small sample and exploratory nature of the study, I wonder if this study would be better framed as a pilot/exploratory study instead?
Author Response

(The authors gave the same response as above.)

Reviewer 3 Report
This article aims to develop a tool to assess the risk of drug therapy errors in hospitals.
The subject is interesting; the study is well conducted; probably reproducible in most developed countries.
The results show that all stages of the medication circuit are important and must be made safe. The tool developed makes it possible to prioritise the actions to be taken.
Risk mapping is carried out in hospitals, in connection with their certification. Calculating criticality is often difficult and subjective; the tool developed helps to objectify this criticality.
A small, anecdotal remark would be to insist on the declaration of events (often underestimated) as well as on feedback on real or possible errors.
Author Response

(The authors gave the same response as above.)

Reviewer 4 Report
The prevention of medication errors is an issue of paramount importance. The tool presented in this manuscript can contribute to this objective, even thought the results obtained are already known.
There are some things that should be improved in order to make more understanding the manuscript.
The most important is that the authors should give more information about PHARIPH. How ha been created? Have the authors previous publications relating to it? In that case they should be referenced. If not, they should explain in detail as part of the present manuscript how the tool has been developed.
As the study is focussed on the error risk of pharmacists’ I think that all concerning the errors derived of patient self-administration should not be considered in this study
Author Response

(The authors gave the same response as above.)

Round 2
Reviewer 1 Report
Thank you for addressing all my comments. In a few places, I could still argue with the authors, but the changes introduced seem sufficient to me.
Reviewer 4 Report
The authors have made important improvements in the manuscript, and answered the aspects asked by the reviewers.